# Red Ginseng Attenuates the Hepatic Cellular Senescence in Aged Mice

**DOI:** 10.3390/biology13010036

**Published:** 2024-01-08

**Authors:** Da-Yeon Lee, Juliana Arndt, Jennifer F. O’Connell, Josephine M. Egan, Yoo Kim

**Affiliations:** 1Department of Nutritional Sciences, Oklahoma State University, Stillwater, OK 74078, USA; dayeon.lee@okstate.edu (D.-Y.L.); juliana.arndt@okstate.edu (J.A.); 2Laboratory of Clinical Investigation, National Institute on Aging, Baltimore, MD 21224, USA; jennifer.oconnell@nih.gov (J.F.O.); eganj@grc.nia.nih.gov (J.M.E.)

**Keywords:** red ginseng, liver, cellular senescence, inflammation, apoptosis, insulin homeostasis

## Abstract

**Simple Summary:**

The aim of this study was to explore the potential impact of red ginseng (RG) on age-associated cellular senescence through in vitro, ex vivo, and 19-month-old aged mouse models. In addition to enhancing insulin homeostasis, RG supplementation modified the canonical cellular senescence pathway and inhibited apoptotic markers. Moreover, RG significantly reduced the senescence-associated inflammatory pathway in the liver. Finally, the RG treatment mitigated cellular senescence in primary hepatocytes and mouse embryonic fibroblasts. In conclusion, these findings suggest that RG has a positive influence on the aging process by regulating hepatic cellular senescence and age-related phenotypes.

**Abstract:**

Cellular senescence is defined as an irreversible cell cycle arrest accompanied by morphological and physiological alterations during aging. Red ginseng (RG), processed from fresh ginseng (*Panax ginseng* C.A. Meyer) with a one-time steaming and drying process, is a well-known beneficial herbal medicine showing antioxidant, anti-inflammatory, and anti-aging properties. The current study aimed to investigate the benefits of RG in alleviating hepatic cellular senescence and its adverse effects in 19-month-old aged mice. We applied two different intervention methods and durations to compare RG’s effects in a time-dependent manner: (1) oral gavage injection for 4 weeks and (2) ad libitum intervention for 14 weeks. We observed that 4-week RG administration was exerted to maintain insulin homeostasis against developing age-associated insulin insensitivity and suppressed cellular senescence pathway in the liver and primary hepatocytes. Moreover, with remarkable improvement of insulin homeostasis, 14-week RG supplementation downregulated the activation of c-Jun N-terminal kinase (JNK) and its downstream transcriptional factor nuclear factor-κB (NF-κB) in aged mice. Lastly, RG treatment significantly reduced the senescence-associated β-galactosidase (SA-β-gal)-positive cells in primary hepatocytes and ionizing radiation (IR)-exposed mouse embryonic fibroblasts (MEFs). Taken together, we suggest that RG can be a promising candidate for a senolytic substance by preventing hepatic cellular senescence.

## 1. Introduction

Cellular senescence is a stable and permanent cell cycle arrest occurring in cells and metabolic organs in response to intrinsic and extrinsic stressors. It is recognized as a major contributor to organismal aging and age-related chronic diseases [1]. Senescent cells exhibit unique characteristics, including the increased activity of senescence-associated β-galactosidase (SA-β-gal), secretion of senescence-associated secretory phenotypes (SASPs), and elevated levels of cell cycle regulatory proteins, which are all essential for confirming cellular senescence [2]. It is well known that either one or both p53/p21^WAF1/CIP1^ and p16^INK4A^/Rb tumor suppressor pathways play crucial roles in modulating the cell cycle in senescent cells [3]. Specifically, the cell cycle regulatory protein p53 triggers apoptosis, a programmed cell death, along with the p53-dependent effector p21 expressions by binding apoptotic agents [4]. Despite senescent cells’ resistance to apoptosis, the aging process frequently leads to excessive hepatocyte apoptosis, causing liver failure and age-associated hepatic disorders [5]. Therefore, the ablation of senescent cells presents a therapeutic strategy against aging and age-related pathologies.

Following these approaches, small-molecule agents with selective clearance of senescent cells, known as ‘senolytics’, have been developed and examined in preclinical and clinical trials to extend health span [6]. However, some of these synthesized drugs still have the potential for mild or severe side effects, prompting increased attention to bioactive compounds from natural food resources as new candidates for senolytics [7,8]. Ginseng (*Panax ginseng* C.A. Meyer), a root from the perennial plant *Panax*, is one of the most important herbal medicines traditionally used in East Asian countries for several millennia [9]. Fresh ginseng is processed into three different types—white, red, and black ginseng—through repetitive steaming and drying processes of up to nine cycles to enhance preservation and efficacy [10]. Among them, red ginseng (RG) is produced from white ginseng by a one-time steaming and drying process, generating RG-specific compounds such as ginsenosides Rg1 and Rb1 [11]. The pharmacological health benefits of RG have been widely investigated, including antioxidant, anti-obesity, anti-inflammatory, neuroprotective, and anti-viral properties in in vitro and in vivo models [12,13,14,15,16]. Yet, whether RG could regulate the age-related specific mechanisms is not fully understood.

In this study, we hypothesized that RG could attenuate hepatic cellular senescence and modulate hepatic apoptosis in the aged condition. Therefore, we assessed the regulatory effects of RG on age-associated cellular senescence and translational markers in cellular and aged animal models, respectively.

## 2. Materials and Methods

### 2.1. Sample Preparation

Red ginseng (RG) extracts for both oral gavage injection and ad libitum dietary intervention studies were kindly obtained from CJ CheilJedang Corporation (Suwon-si, Gyeonggi-do, Republic of Korea) [17] and BTC Corporation (Ansan-si, Gyeonggi-do, Republic of Korea) [18], respectively.

### 2.2. Animals

In the investigation involving oral gavage interventions, young male C57BL/6J mice (9 weeks old) and aged male C57BL/6 mice (19 months old) were sourced from the Jackson Laboratory (Bar Harbor, ME, USA) and the National Institute on Aging (NIA) Aged Rodent Colony at Charles River Laboratories (Frederick, MD, USA). The study, conducted at the NIA and accredited by the American Association for Accreditation of Laboratory Animal Care (AAALAC), received approval from the Animal Care and Use Committee (ACUC) of the NIA Intramural Program. Upon arrival at the NIA facility, mice underwent a one-week acclimation period with a standard NIH chow diet (Teklad Global Rodent Diet Envigo, Indianapolis, IN, USA) and ad libitum access to water. Subsequently, the animals were randomized into three groups: 9-week-old mice receiving orally administered distilled water (Young), 19-month-old mice receiving orally administered distilled water (Old), and 19-month-old mice receiving 300 mg/kg RG in distilled water (Old + RG) (n = 6 per group). The RG dosage was determined based on a prior study [14]. Over a four-week period, oral gavage injections were administered every four days, with regular monitoring of body weight and food intake. At the conclusion of this timeframe, mice were euthanized, and liver tissues were collected for further analyses.

In the ad libitum dietary intervention study, twenty 19-month-old male C57BL/6 mice were obtained from the NIA Aged Rodent Colony at Charles River Laboratories, transferred to the Oklahoma State University Animal Resources Facility, and acclimated for one week. Initial assessments included measurements of body weight and body composition. The mice were then randomly assigned to two groups: a normal chow diet (Control) and Control with 0.4% (*w*/*w*) RG (RG) (n = 10 per group). Body weight and food intake were monitored weekly over the 14-week intervention period.

### 2.3. Glucose and Insulin Tolerance Test

In the glucose tolerance assessment (GTT), mice that had undergone a 16-h fast were subjected to blood glucose level measurements at 0 min, 15, 30, 60, 90, and 120 min following an intraperitoneal (IP) injection of 2 g/kg glucose, prepared as a 50% (*w*/*w*) glucose solution (Alpha Teknova Inc., Hollister, CA, USA). The insulin tolerance test (ITT) involved the measurement of blood glucose levels in mice that had fasted for 6 h. Measurements were taken at 0 min, 15, 30, 60, 90, and 120 min after IP administration of 0.5 or 1 IU/kg recombinant human insulin (Novo Nordisk Inc., Plainsboro, NJ, USA). Blood glucose levels were determined using a handheld glucometer (CONTOUR^®^ NEXT EZ; Ascensia Diabetes Care, Parsippany, NJ, USA).

### 2.4. Immunoblotting Assay

The primary hepatocytes and mouse liver tissues were subjected to homogenization using RIPA Lysis and Extraction Buffer (for hepatocytes) and T-PER™ Tissue Protein Extraction Reagent (for liver tissues) from Thermo Fisher Scientific (Waltham, MA, USA). The homogenization process utilized the OMNI Bead Ruptor 24 (Omni-Inc., Kennesaw, GA, USA). To isolate nuclear and cytoplasmic fractions from the liver tissues, NE-PER Nuclear and Cytoplasmic Extraction Reagents were employed as per the manufacturer’s instructions (Thermo Fisher Scientific). Additional supplementation of homogenates with PhosSTOP™ phosphatase inhibitor and cOmplete™ Mini Protease Inhibitor Cocktail (Sigma–Aldrich, St. Louis, MO, USA) was performed. Determination of the protein concentration was carried out using the bicinchoninic acid (BCA) assay kit from Thermo Fisher Scientific. After homogenization, samples, ranging in protein loading from 45 to 120 μg, underwent SDS-PAGE resolution under reducing conditions, followed by transfer to polyvinylidene fluoride (PVDF) membranes. Subsequent blocking of membranes with a blocking reagent (LI-COR, Lincoln, NE, USA) at room temperature for one hour was conducted. The membranes were then incubated overnight at 4 °C with primary antibodies, which included cleaved caspase-3, caspase-3, p53, p21, c-Jun N-terminal kinase (JNK) 1/2, p65, β-actin, β-tubulin, and histone deacetylase (HDAC) 1 from Cell Signaling Technology (Danvers, MA, USA), as well as p16 and p15 from Santa Cruz Biotechnology, Inc. (Dallas, TX, USA). After washing with tris-based saline with Tween 20 (TBS-T), appropriate secondary antibodies (anti-rabbit IgG from Cell Signaling Technology or anti-mouse IgG from Santa Cruz Biotechnology, Inc.) were applied in 5% non-fat dry milk for one hour. Following three TBS-T washes, membranes were developed using a chemiluminescence assay system from Thermo Fisher Scientific. A visualization of bands on the membranes occurred through autoradiographic X-ray films from Thomas Scientific (Swedesboro, NJ, USA). A scanning of the Western blot images and saving as Tiff files was followed by inversion and a subsequent assessment of integrated density using ImageJ software (NIH). Normalization of phosphorylated protein levels was then performed relative to their corresponding total protein levels. The original immunoblots are presented in Appendix A.

### 2.5. Isolation of Primary Hepatocytes and Mouse Embryonic Fibroblasts (MEFs)

Primary hepatocytes were obtained from mice in both the Control and RG-fed groups (n = 2 per group) using a two-step collagenase perfusion technique, as outlined in a prior study [19]. The isolated hepatocytes were then cultured in Waymouth’s medium, supplemented with 10% fetal bovine serum (FBS), 1% penicillin/streptomycin (P/S), 1 nM of insulin, and 1 uM of dexamethasone. Simultaneously, primary mouse embryonic fibroblasts (MEFs) were derived from E13.5 embryos of 9-week-old C57BL/6J female mice (n = 3). Following isolation, MEFs were cultured in high-glucose Dulbecco’s Modified Eagle Medium (DMEM), enriched with 20% FBS, 1% P/S, 2 mM L-glutamine, 1X non-essential amino acids, and 20 mM HEPES (4-(2-hydroxyethyl)-1-piperazineethanesulfonic acid) from Thermo Fisher Scientific.

### 2.6. Cellular Senescence Induction in MEFs by Ionizing Radiation (IR)

MEFs were seeded onto 60-mm culture plates and cultured until they achieved a confluence of 70–80%. Subsequently, they underwent treatment with 5 μg/mL of aqueous RG extract for a duration of 7 days, with changes in the culture media every 2 days [20]. To determine the optimal RG concentration, a dose-dependent treatment was conducted following the protocol outlined in a prior report [20]. For the acute irradiation phase, the ionizing radiation dose was selected based on a previous study. MEF cells, both with and without RG treatment, were exposed to a 20 Gy dose of γ-ray using a 137Cs source (Gammacell^®^ 40 Exactor; Best Theratronics Ltd., Kanata, ON, Canada) [20].

### 2.7. Senescence-Associated β-Galactosidase (SA-β-gal) Staining

The primary hepatocytes and irradiated MEFs were subjected to SA-β-gal staining using a Senescence β-Galactosidase Staining Kit (Cell Signaling Technology), following the manufacturer’s guidelines. Quantification of the percentage of SA-β-gal-positive cells included the enumeration of both positive and total cells from the images, with a sample size ranging from n = 35 to 38.

### 2.8. Statistical Analyses

Experimental data underwent the analysis using GraphPad Prism 9 software (GraphPad Software, San Diego, CA, USA). An ordinary two-way repeated-measure analysis of variance (ANOVA) was utilized to evaluate variables, including body weights, accumulated food intakes, 6-h fasting blood glucose levels, GTT, and ITT, followed by Tukey’s multiple comparison tests. The presentation of quantitative data follows the format of mean ± standard error of the mean (SEM). For the assessment of the area under the curve (AUC) and Western blot band density, a one-way ANOVA was conducted, and subsequently, either Tukey’s multiple comparison test or Student’s *t*-test was applied after the outlier test (α = 0.05).

## 3. Results

### 3.1. Red Ginseng Supplementation Is Involved in Glucose Homeostasis in Aged Mice

We initially examined the impact of short-term dietary RG on age-related symptoms, including changes in body weight and the dysregulation of glucose and insulin homeostasis in naturally aged mice. Throughout the 4 weeks of oral gavage administrations, no significant changes in body weight or accumulated food intake were observed between both the Control and RG groups (Figure 1A,B). Subsequently, we assessed the 6-h fasting blood glucose levels before and after the intervention. Remarkably, RG decreased the fasting blood glucose level after 4 weeks from 144.83 to 105.50 mg/dL (*p* < 0.01) (Figure 1C). However, there was no significant difference in blood glucose levels after a 16-h fasting condition at week 4 between the two groups (Figure 1D). GTT results showed that there was no difference between the Control and RG groups (Figure 1E). An ITT was performed at week 4, and we observed that the RG supplementation showed a trend of improved insulin sensitivity, although it did not reach significance (Figure 1F). Taken together, these data suggest that RG supplementation for 4 weeks provides a potential effect on the regulation of glucose and insulin homeostasis in the aged condition.

### 3.2. Red Ginseng Attenuates Hepatic Cellular Senescence Pathways in Aged Mice

Given the potential of RG to influence glucose and insulin homeostasis in aged mice, we investigated its impact on aging and the associated mechanisms. The tumor suppressor proteins p53 and p53-dependent p21 are key effectors that modulate cellular senescence and organismal aging [21]. Elevated expressions of cyclin-dependent kinase (CDK) inhibitors p16 and p15 in metabolic organs are featured characteristics of senescent cells [22]. Therefore, we first assessed the protein expression levels of p53, p21, p16, and p15 in the liver of mice after 4 weeks of RG intervention. RG supplementation not only significantly suppressed the expression of p21 (*p* < 0.01) and p15 (*p* < 0.05), but also decreased p53 and p16 expression compared to the Old group (Figure 2A). Next, we isolated primary hepatocytes, the mesenchymal cells that regulate vital metabolisms in the liver, to confirm these findings from whole liver tissues. We observed decreased p15 expression levels in primary hepatocytes from the RG-fed mice (Figure 2B), indicating that RG supplementation downregulates hepatic cellular senescence stimulators. To further understand how RG alleviates the hepatic cellular senescence, we analyzed the expression levels of apoptosis markers in hepatocytes. The ratio of cleaved caspase-3 to caspase-3 was significantly decreased by RG administration (*p* < 0.05) (Figure 2C). Collectively, RG was shown to contribute to the suppression of cellular senescence by regulating cell cycle modulators in the liver.

### 3.3. Red Ginseng Administration Improves Insulin Homeostasis in Aged Mice

While oral gavage efficiently delivers targeted substances directly into the murine stomach, it can induce physiologic stress, especially with mid- and long-term intervention studies in experimental animals [23]. Therefore, we conducted an RG intervention study using ad libitum feeding, a more physiologically relevant approach, with an extended duration of 14 weeks to evaluate RG’s time-dependent effects. Consistent with the short-term oral gavage injection for 4 weeks (Figure 1A,B), the 14-week RG intervention did not result in changes in body weight (Figure 3A) or accumulated food intake (Figure 3B) compared to the Control group. Blood glucose levels following 6-h fasting were significantly increased at week 7 in the Control group compared to the baseline 120.22 to 167.33 mg/dL (*p* < 0.0001), whereas RG treatment successfully maintained 6-h fasting blood glucose levels compared to week 0 from 129.67 to 134.00 mg/dL (Figure 3C). Similar to the results of the 4-week RG intervention illustrated in Figure 1E, the GTT at week 8 showed no difference between the Control and RG-fed groups (Figure 3D). Intriguingly, ITT results at week 7 demonstrated that RG supplementation significantly improved insulin sensitivity in the RG-fed mice (Figure 3E). These results support the finding that extended RG supplementation provides more health benefits in metabolic changes, particularly in insulin and glucose homeostasis, compared to relatively short-term interventions (4 weeks) in aged mice.

### 3.4. Red Ginseng Suppresses the JNK/NF-κB Signaling Pathway in Aged Mice

c-Jun N-terminal kinase (JNK), one of the three compartments of mitogen-activated protein kinases (MAPKs), has been implicated in the regulation of senescence traits and the induction of cellular apoptosis [24]. Thus, we examined the effect of RG supplementation on JNK activation. RG remarkably suppressed the expression levels of activated JNK, resulting in a significant decrease in the ratio of p-JNK to JNK protein expression levels with RG treatment (*p* < 0.05) (Figure 4A). To validate these results, we subsequently assessed the expression levels of the p65 subunit of nuclear factor-κB (NF-κB), a downstream transcriptional factor of JNK that plays a pivotal role in immunosenescence and apoptosis regulated by the inhibitory IκBα protein [25]. We fractionated proteins from the nuclear and cytoplasmic compartments in the liver tissues and measured p65 protein expression levels in both fractions, since p65 is a transcriptional factor translocated into the nucleus. We observed that RG supplementation significantly inhibited p65 expression in both nuclear and cytoplasmic fractions, suggesting the role of RG in suppressing activated NF-κB as a protector against apoptosis in the aged mice (*p* < 0.05) (Figure 4B).

### 3.5. Red Ginseng Alleviates Cellular Senescence in Primary Hepatocytes and MEFs

The aforementioned observations prompted us to investigate whether RG could serve as a senolytic to ameliorate cellular senescence. We first isolated primary hepatocytes from both the Control and RG-fed aged mice and identified senescent cells using a SA-β-gal staining assay. Notably, there was a significant reduction in the percentage of SA-β-gal-positive hepatocytes from RG-fed mice (*p* < 0.0001) (Figure 5A). Next, we utilized primary MEFs, a commonly used in vitro model known to readily undergo senescence within a short period [26]. We exposed these MEFs to acute IR exposure to induce cellular senescence. Consistent with the observations from the previous ex vivo results with primary hepatocytes, senescence in MEFs cultured in growth media with 5 μg/mL of RG was delayed compared to non-RG-treated irradiated MEFs (Figure 5B). These findings suggest that RG may play a role in regulating cellular senescence by reducing the number and density of senescent cells.

## 4. Discussion

As one of the hallmarks of aging, cellular senescence deteriorates molecular functions in multiple organs, leading to cellular proliferation arrest and contributing to several age-related pathologies [27]. This connection has spurred scientific research into bioactive agents, known as senolytics, with a focus on natural food resources. Previous studies have reported the preventive effects of RG on cellular senescence in skeletal muscle, skin, and vascules [28,29,30]. However, the impact of RG on cellular senescence in the aged liver remains unclear.

In this study, we employed two different RG interventional regimens: (1) 4 weeks oral gavage (short-term) and (2) 14 weeks ad libitum (mid-term). While the 4-week RG intervention showed mild metabolic changes, the 14-week intervention study demonstrated a more pronounced improvement in age-associated impaired insulin sensitivity. This phenotypic evidence supports the potential positive effect of RG on the duration of consumption, particularly in relation to age.

In the aging process, the canonical cellular senescence pathway is initiated by two conserved and interactive pathways—p53/p21^CIP1/WAF1^ and p16^INK4a^/Rb cascades—by arresting the G0/G1 phase and transitioning from G1 to S phase of the cell cycle, respectively [3,31,32]. Alternatively, the CDK4/6 inhibitor p15^INK4b^, mediated by the transforming growth factor (TGF)-β, is highly expressed in aged hepatocytes, leading to inhibited cellular proliferation and increased apoptosis [33]. The accumulation of these irreversible epigenetic alterations in metabolic organs, particularly in the liver, exacerbates age-related pathophysiological conditions, such as type 2 diabetes, cardiovascular disease, and neurodegenerative disorders [34]. In this study, we observed that 4 weeks of RG oral administration downregulated the hepatic cellular senescence pathway through a marked decrease in canonical p21 and non-canonical p15 expression levels, followed by the inhibition of apoptotic markers. Previous studies have shown that ginsenoside Rb1, mainly distributed in RG, regulates the p53-p21-CDK2 axis and decreases caspase-3 activation in cardiac aging [35]. Here, we suggest that bioactive compounds in RG, such as ginsenoside Rb1, might act as key components to delay hepatic cellular senescence in the aged liver, contributing to the prevention of senescence in the liver.

The regulatory role of JNK is well-documented, encompassing not only cellular senescence but also autophagy [36]. Furthermore, in response to intracellular and extracellular stresses, the activations of JNK and NF-κB induce insulin resistance through the serine phosphorylation of the insulin receptor substrate (IRS)-1 or IRS-2, subsequently blocking insulin signaling. This suggests the JNK/NF-κB pathway as an important therapeutic target for lifespan extension and insulin homeostasis [37,38,39]. In this study, alongside improved insulin tolerance, we observed that RG supplementations for 14 weeks significantly suppressed the phosphorylation of hepatic JNK and NF-κB signaling in naturally aged conditions. Notably, the RG treatment attenuated age-induced cellular senescence in primary hepatocytes and MEFs. These findings align with a previous report indicating that aqueous RG extracts suppressed inflammatory translational markers, augmented autophagy activities, and downregulated mRNA expressions of SASPs in aged mouse models [40]. This is likely attributed to 20(R)-ginsenoside Rg3, a rare ginsenoside abundant in RG, which exerts strong anti-inflammatory and antioxidant capacity against hepatic oxidative stress and apoptosis [41,42].

One limitation of this study is that we did not analyze the bioactive components in RG, and thus, we could not identify the essential ginsenosides alleviating hepatic cellular senescence. While this points towards a need for further studies to profile and identify key ginsenosides on cellular senescence in RG, our study provides evidence that using a whole RG extract might be more practical than relying on single compounds, inducing synergistic effects among other components in RG. Additionally, our animal experiments involved two different intervention durations: 4 weeks (short-term) and 14 weeks (mid-term). However, further studies are necessary to determine whether RG exerts the same senescence-preventive effect with administrations longer than 6 months (long-term). Given that no significant phenotypic and toxicological changes were observed through long-term administration of RG (up to 12 months) [43], an extended RG intervention in an aged mouse model would provide more information about the optimal duration for RG supplementation to alleviate hepatic cellular senescence.

## 5. Conclusions

In summary, our research demonstrated that dietary supplementation with RG suppressed the p53-p21/p16 cascade, regulated the apoptotic pathway, and downregulated the JNK/NF-κB signaling pathways in in vitro, ex vivo, and in vivo models. In conclusion, this study suggests that RG could be a potential candidate for a senolytic substance to attenuate age-associated hepatic cellular senescence.

## Figures and Tables

**Figure 1 biology-13-00036-f001:**
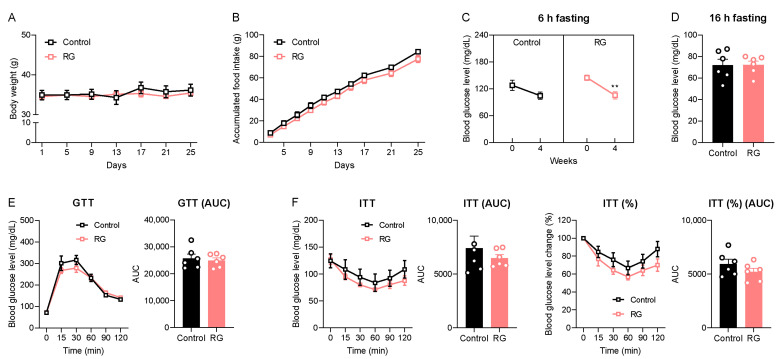
Red ginseng supplementation for 4 weeks expresses the possibility of being a senolytic in aged mice. The 19-month-old male C57BL/6 mice were orally administered distilled water (Control, n = 6) or 300 mg/kg of aqueous RG extract (RG, n = 6) for 4 weeks. (**A**) Body weight (g) and (**B**) accumulated food intake (g) were monitored every 4 days. (**C**) 6 h fasted blood glucose levels (mg/dL) on weeks 0 and 4. (**D**) 16 h fasted blood glucose levels (mg/dL) on week 4. (**E**) Glucose tolerance test (GTT; 2 g/kg) on week 4 and the calculation of area under the curve (AUC). (**F**) Insulin tolerance test (ITT; 1 IU/kg) on week 4 and AUC. The GTT and ITT were conducted at three-day intervals. Ordinary two-way ANOVA was used to analyze body weights, food intakes, GTT, and ITT, followed by Student’s *t*-tests for blood glucose levels and AUC. The results are expressed as mean ± SEM. (** *p* < 0.01).

**Figure 2 biology-13-00036-f002:**
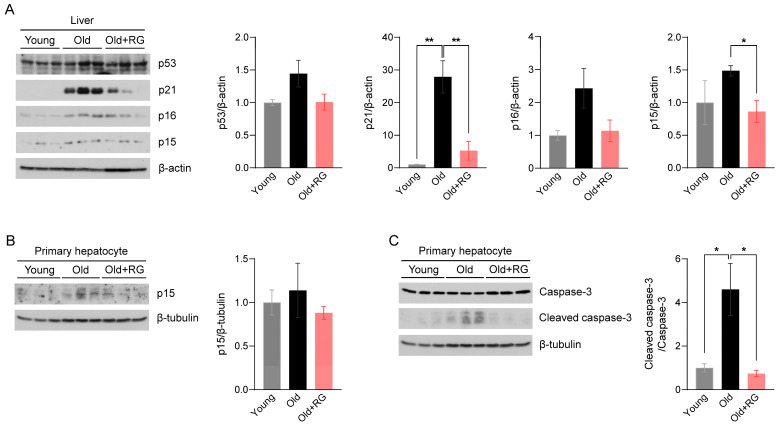
Red ginseng administration for 4 weeks suppresses the hepatic cellular senescence pathway in aged mice. Distilled water was orally administered to 9-week-old (Young) and 19-month-old (Old) mice, and 300 mg/kg RG in distilled water was given to 19-month-old (Old + RG) mice (n = 6 per group) for 4 weeks. (**A**) Total liver tissue lysates were immunoblotted with cellular senescence-related markers: p53, p21, p16, and p15. (**B**,**C**) Primary hepatocytes were isolated from Young, Old, and Old + RG mice, followed by immunoblotting with (**B**) p15 and (**C**) cleaved caspase-3 and caspase-3. Ordinary two-way ANOVA was used for the quantification of immunoblots followed by Tukey’s multiple comparison tests. The data are expressed as mean ± SEM (* *p* < 0.05, ** *p* < 0.01).

**Figure 3 biology-13-00036-f003:**
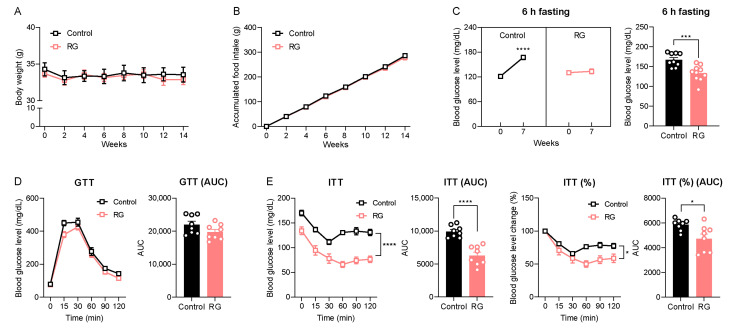
Red ginseng intake for 14 weeks maintains insulin homeostasis in aged mice. 19-month-old male C57BL/6 mice were fed ad libitum with a normal chow diet (Control; n = 10) or 0.4% (*w*/*w*) RG (RG, n = 10) for 14 weeks. (**A**) Body weight (g) and (**B**) accumulated food intake (g) was assessed every two weeks. (**C**) 6 h fasted blood glucose levels (mg/dL) on weeks 0 and 7. (**D**) Glucose tolerance test (GTT; 2 g/kg) on week 8 and the calculation of area under the curve (AUC). (**E**) Insulin tolerance test (ITT; 0.5 IU/kg) on week 7 and the calculation of AUC. Ordinary two-way ANOVA was used to analyze body weights, food intake, GTT, and ITT. Student’s *t*-test was used to analyze blood glucose levels and AUC. All results are expressed as mean ± SEM. (* *p* < 0.05, *** *p* < 0.005, **** *p* < 0.001).

**Figure 4 biology-13-00036-f004:**
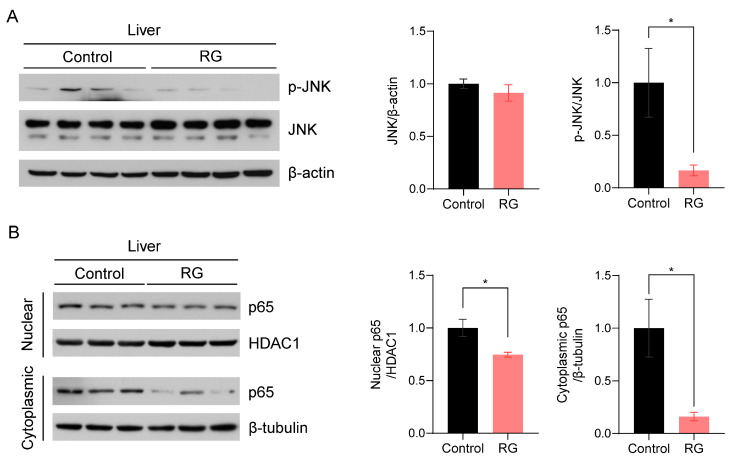
Red ginseng uptake for 14 weeks downregulates the hepatic JNK/NF-kB signaling pathway in aged mice. Liver tissues were collected from Control and RG-fed mice after 14-week interventions and immunoblotted with inflammation-related antibodies. (**A**) The immunoblots were quantified by normalizing p-JNK and JNK to β-actin and p-JNK to total JNK. (**B**) The nuclear and cytoplasmic fractions were fractionated from the mouse liver tissues. The quantification of immunoblots was performed by the normalization of nuclear p65 to HDAC1, and cytoplasmic p65 to β-tubulin. Student’s *t*-test was used to quantify the immunoblots. The results are presented as mean ± SEM. (* *p* < 0.05).

**Figure 5 biology-13-00036-f005:**
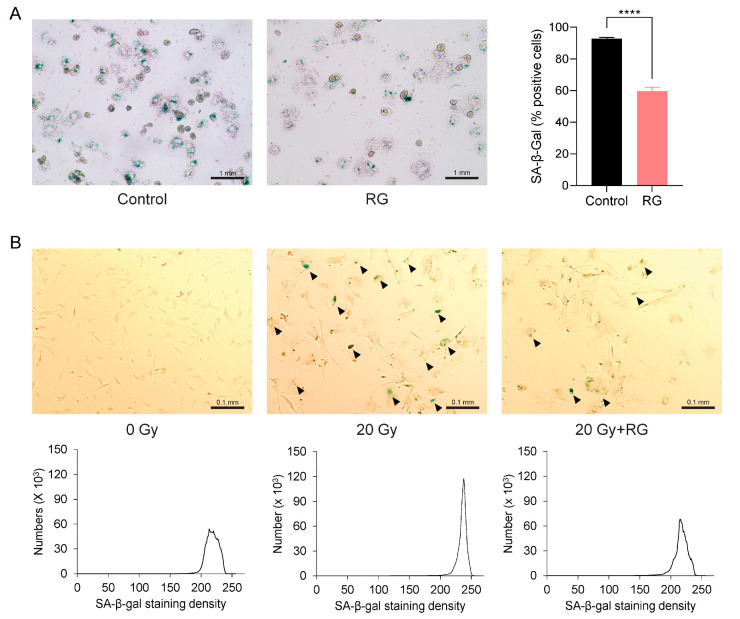
Red ginseng delays cellular senescence in primary hepatocytes from aged mice and primary mouse embryonic fibroblasts (MEFs). Representative morphologies of senescence-associated β-galactosidase (SA-β-gal) stained (bluish green) cells. (**A**) Primary hepatocytes isolated from Control and RG-fed mice after 14 weeks of intervention (**left**) and SA-β-gal quantification plot (**right**) (scale bar = 1 mm). (**B**) Primary mouse embryonic fibroblasts (MEFs) were applied with the acute ionizing radiation of 0 Gy, 20 Gy, or 20 G with 5 μg/mL of aqueous RG. Black arrows point to SA-β-gal stained MEFs (**upper**). SA-β-gal quantification plots based on staining density (**below**) (scale bar = 0.1 mm). Student’s *t*-test was used to analyze the percentage of SA-β-gal-positive cells. Results are expressed as mean ± SEM (**** *p* < 0.001).

## Data Availability

The datasets used and/or analyzed during the current study are available from the corresponding author on reasonable request.

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
