# Peer review of "Red Ginseng Attenuates the Hepatic Cellular Senescence in Aged Mice"

_biology, 2024, doi:10.3390/biology13010036_

Round 1

Reviewer 1 Report

Comments and Suggestions for Authors

In this study Lee and colleagues evaluated the senolytic properties of the supplementation of red ginseng (RG) on young and aged mice. Overall, the subject is an interesting one, with positive evidence on the supplementation of ginseng accumulating over the years. The study seems to have been well conducted, the results seem to have been correctly analyzed and discussed. The authors observed a considerable effect of RG in the metabolic (glucose and insulin) profile of mice, although the most expressive results were the attenuation of inflammatory and senescent pathways in liver cells and embryonic fibroblasts.

This reviewer has the following issues:

-        Even though the chemical characterization of RG could not be performed, at least the authors should provide the most information possible of this sample – type of solvent, preparation type, part of the plant used for this preparation, etc. Even if such information cannot be provided, at least the authors should provide with an overall description of what compounds were expected, considering the known composition from previous studies;

-        The limitations should be presented in a specific section, not in the body of text of the Discussion;

-        How did the authors calculate the sample size for these animal groups?

-        How many days difference was between GTT and ITT? This should be stated in the text;

-        Considering that no animals were obese and that no high fat diets were used, how can the authors explain (Figure 1) the 6h fasting glucose levels of around 120 mg/dL in the aged group? For this fasting period, these values are compatible with a prediabetic state. Was this prediabetic state intended by the authors? If so, it should be explained in the text.

Reviewer 2 Report

Comments and Suggestions for Authors

Lee et al. reported that red ginseng attenuates the hepatic cellular senescence in aged mice. Furthermore, the potential mechanism was also evaluated. The manuscript was well prepared. Thus, this work is suitable for publication after some modifications.

1.     Why the authors used the two-way method for statistical analyses?

2.     The composition of Red ginseng (RG) extracts is unknown.

3.     The JNK/NF-κB signaling pathway is very complex, for example, IκBα, phospho-IκBα are also important proteins in this pathway.

4.     Please check the statistical analyses in Fig. 2.

Reviewer 3 Report

Comments and Suggestions for Authors

Firstly, I would like to congratulate the authors for the quality of the manuscript entitled “Red Ginseng Attenuates the Hepatic Cellular Senescence in Aged Mice”. The manuscript indicates the potential of red ginseng as a potential agent capable of attenuating age-associated hepatic cellular senescence. The experiments were conducted appropriately and support the conclusions. Also, the limitations of the study were well defined.  I would like to congratulate the authors for the work.  However, a few clarifications are required to contribute with the manuscript, as follow:

Line 120: Indicate the protein concentration of the samples.

Line 170: I would remove ‘and insulin’ since you could not observe changes statistically significant in the regulation of insulin homeostasis.

Figures 2 and 4: Each sample on the western blot were applied in a triplicate? I suggest to make it clear.

Figures 2A: In the sample ‘Old + RG’, there’s a clear difference in p21 expression. Why?
